# Phase Composition Effects on Dynamic Behavior and Strain Rate Sensitivity in Metastable β-Ti Alloys

**DOI:** 10.3390/ma15124068

**Published:** 2022-06-08

**Authors:** Tao Wang, Yong Feng, Xianghong Liu, Kaixuan Wang, Shaoqiang Li, Feng Zhao

**Affiliations:** 1State Key Laboratory of Solidification Processing, Northwestern Polytechnical University, Xi’an 710072, China; wang-tao@c-wst.com (T.W.); yfeng@c-wst.com (Y.F.); xhliu@c-wst.com (X.L.); 2Western Superconducting Technologies Co., Ltd., Shaanxi Province Engineering Laboratory for Aerial Material, Xi’an 710018, China; kingsin@c-wst.com (K.W.); sqli_wst@163.com (S.L.); 3Institute for Advanced Study, Chengdu University, Chengdu 610106, China

**Keywords:** metastable β-Ti alloys, stain rate sensitivity, mechanical behavior, phase composition

## Abstract

In this study, high strain rate tension tests are conducted to determine and compare the dynamic mechanical behaviors and deformation mechanisms of different phase composition α-β metastable β-Ti alloys using a split Hopkinson tension bar. Two typical bimodal equiaxed α_p_ + β and lamellar α_s_ + β Ti-45551 alloy microstructures are formed through different hot working and thermal processing for investigating the effect of phase composition or microstructure on mechanical properties and strain rate sensitivity. It is demonstrated that dislocation nucleation and motion in the α/β phase and dislocation tangle or pile up at the α/β interface are typical deformation modes in both of the typical dual-phase Ti alloys at quasi-static loading conditions. Under dynamic loading, both the strength and ductility show a clearly positive strain rate dependence, which is directly related to dislocation activation in the α + β Ti-45551 alloy. Based on microstructure characterizations, it is shown that deformation twinning starts to become a major deformation mechanism in equiaxed α_p_ + β microstructures under dynamic loading conditions. However, deformation twins are not favored in the lamellar α_s_ + β Ti-45551 alloy due to its nano phase size. Finally, the mechanical behaviors and strain rate sensitivity are strongly dependent on the phase composition of metastable β-Ti alloys.

## 1. Introduction

Because of the high strength to weight ratio, the broad range of service temperatures, and the excellent corrosion resistance, metastable β-Ti alloys have widespread application in the aviation, aerospace, and automotive industries [1,2,3]. Generally, the microstructure of metastable β-Ti alloys consists of α and β phases. Generally, the high strength always is attributed to the Hexagonal close-packed (HCP) α phase, and the ductility comes from the Body-centered cubic (BCC) β phase. Thus, the superior strength–ductility synergy and microstructure tailoring method are governed by the α and β phase composition of metastable β-Ti alloys [4].

By means of thermo-mechanical processing routes, many kinds of microstructures can be produced, including equiaxed α_p_ + β, lamellar α_s_ + β, and equiaxed α_p_ + lamellar α_s_ + β bimodal microstructures. As a result, the phase composition or microstructure heterogeneity exhibits a great effect on the mechanical behaviors of these metastable β-Ti alloys. Many efforts have been proposed to describe the relationship between mechanical properties, microstructural evolution, and deformation mechanisms in the metastable β-Ti alloys. Generally, a high density of dislocation tends to pile up against the α/β interfaces during plastic deformation under quasi-static loading conditions [5]. Recently, based on studies of the α phase size effect on deformation modes of the Ti-5553 alloy, it has been shown that dislocation spacing and dislocation type are controlled by the size and distribution of the α phase [6,7]. Moreover, with the decreasing of lamellar α_s_ thickness, the strength of lamellar α_s_ + β-Ti alloys increases following a Hall–Petch relationship [8,9]. The phase size, volume fraction, morphology, and distribution of the α phase are essential microstructure parameters which govern the deformation modes and/or mechanical behaviors [10]. As a result, for improving the strength and ductility, it is necessary to understand the phase composition effects on mechanical behaviors during plastic deformation.

For structural engineering applications, where high strength metastable β-Ti alloys are always subjected to high strain rate loading conditions, the role and effect of phase composition on mechanical properties must be understood. However, the focus of prior studies on mechanical behaviors of metastable β-Ti alloys was primarily under quasi-static loading conditions. Generally, these alloys have shown a strong strain rate sensitivity (SRS) at ambient temperature and low strain rates [11,12,13]. Few studies have been conducted on the dynamic deformation behaviors of metastable β-Ti alloys. Therefore, a systematic study of the phase composition on the dynamic mechanical behaviors and plastic deformation of metastable β-Ti alloys is required.

The present work aimed to investigate the difference in the mechanical behaviors of a metastable β-Ti alloy under dynamic loading at a range of strain rates (1 × 10^−3^–4 × 10^3^ s^−1^). Moreover, the effects of phase composition on the mechanical behaviors and plastic deformation under dynamic tensile loading were investigated. In this work, a series of quasi-static and dynamic tensile tests was conducted on a metastable β-Ti alloy at room temperature. The mechanical behaviors and deformation mechanisms were carefully investigated.

## 2. Experimental Procedure

### 2.1. Materials

In this work, the as-received material was Ti-45551 (mass: Al, 4.00%; Mo, 4.99%; V, 5.03%; Cr, 5.01%; Nb, 1.03%; Ti, balance). The Ti-45551 alloy ingot was first fabricated by cold crucible induction levitation melting. The β-transus temperature was about 800 °C. The β-field forging and the multi-axial α + β field forging were conducted by hydraulic press. The Ti-45551 alloy was solution treated (ST) above β-transus temperature (870 °C) for 2 h followed by air cooling (AC), and then aging at 525 °C for 4 h and AC. Scanning electron microscope and electron back scattered diffraction (EBSD) were used to examine the microstructure of the Ti-45551 alloy. Figure 1a shows the typical α + β bimodal phase microstructure. Figure 1b illustrates the inverse pole figure and phase-mapping of the ST and aged Ti-45551 alloy. The resulting microstructure included about 24% equiaxed primary α_p_ phase (grain size ~ 2 μm) and equiaxed β phase grain (~40 μm). In order to get an additional insight into the microstructure, Figure 2a,b shows that the applied heated treatment resulted in the microstructure of a primary α_p_ and α_s_ lamellar microstructure.

In order to obtain a uniform lamellar α_s_ phase, equiaxed α_p_ + β samples were then aged at 360 °C for 8 h and AC, followed by another aging at 550 °C for 8 h. The microstructure of the two-stage aged Ti-45551 is shown in the Scaning electro microscope (SEM) and bright field Transmission electron microscope (TEM) image in Figure 2c,d. Figure 2c shows the α_s_ lamellar structure evenly lies on the equiaxed β grains. The number density of the lamellar α_s_ phase was very high. The width of the lamellar α_s_ phase was about 30–60 nm, while its length was about 200 nm to 2 μm, as seen in Figure 2d. Moreover, in both the equiaxed α_p_ + α_s_ + β and lamellar α_s_ + β bimodal microstructures, tangled dislocations were found near the interface of α/β.

### 2.2. Mechanical Properties

The quasi-static tensile tests were performed using an Mechanical tests system (MTS) testing system (Sansi, Chengdu, China) with a strain rate of 10^−3^ s^−1^ at room temperature. During the tensile tests, the elongation was recorded by an extensometer with gauge length of 12 mm. As tensile samples break in the working area of the extensometer, the total strain histories are recorded by the testing systems under quasi-static loading.

The dynamic tensile tests were performed at strain rates of (~6 × 10^2^ s^−1^, 2 × 10^3^ s^−1^, and 4 × 10^3^ s^−1^) by using a standard Split hopkinson tension bar (SHTB) system at RT. In order to ensure the repeatability of the tensile test data, at least three tests were conducted at a certain strain rate. Figure 3a shows the schematic illustration of SHTB, which was used in this work. During dynamic loading, an elastic stress wave is generated when the projectile strikes the incident bar. As the stress wave reaches the interface of the incident bar/specimen, it partially propagates through the sample into the transmission bar as a transmitted wave. In the meantime, the incident wave partially reflects back into the incident bar as a reflection wave at the incident bar/specimen interface. The reflected wave and transmitted wave were recorded using strain gages, which were fixed on elastic bars, as seen in Figure 3. During the test, the incident, reflection, and transmission voltage–time signals were recorded by the data acquisition system. According to the one-dimensional stress wave theory, the true stress, true strain, and strain rate of each impact can be calculated as follows:(1)$$\sigma_{S}\left(t\right)=E\left(\frac{A}{A_{S}}\right)\varepsilon_{T}\left(t\right)$$
(2)$$\varepsilon_{S}\left(t\right)=\frac{2C_{0}}{l_{s}}\int^{\;}\begin{array}{c} t\\ 0 \end{array}\varepsilon_{R}\left(t\right)dt$$
(3)$${\dot{\varepsilon }}_{S}\left(t\right)=\frac{2C_{0}}{l_{s}}\varepsilon_{R}\left(t\right)$$
where *ε_R_* is the reflection strain, *ε_T_* is the transmission strain, *E* is Young’s modulus, *A* is the cross-sectional area, *C_0_* is the longitudinal wave speed of the bars, *A_s_* is the cross-sectional area of the specimen, and *l_s_* is the length of the specimen. The details of the SHTB technique can be found elsewhere [14,15].

The precise dimensions of the Ti alloy samples for quasi-static and dynamic tensile tests are shown in Figure 3b respectively.

### 2.3. Microstructure Characterization

In this work, the microstructure characterizations were conducted by SEM (JEOL, Tokyo, Japan), Electron backscatter diffraction EBSD (JEOL), and TEM (JEOL) at different strain rates at room temperature. For the EBSD characterization, the samples were first mechanically grinded with 400–1200 grit SiC paper and then mechanically polished by using 0.2 μm diamond paste. Subsequently, each sample was electrolytically polished to remove surface strain. The EBSD data were obtained by HKL Channel 5 analysis software (Version 5.12, 2019, Oxford instruments, Oxford, UK). The EBSD images were acquired under the following conditions: a 20 kV acceleration voltage and a 0.1 μm scan step size on a rectangular scan grid. After plastic deformation, the TEM samples were taken from the deformed region. The bright field and the high0resolution images of the deformed Ti alloy were acquired by JEOL F200 (JEOL). TEM disk samples were used after mechanical polishing and thinned to perforation using precision ion thinning treatment.

## 3. Result and Discussion

### 3.1. Phase Composition Effects on Mechanical Behaviours

Generally, tensile properties are affected by the microstructure, such as the phase volume fraction, phase size, morphology of equiaxed α_p_ and lamellar α_s_, and transformed β phase. Based on the two typical phase compositions or microstructures (α_p_,_s_ + β, α_s_ + β) of Ti-45551 alloy, the effect of microstructure on mechanical behaviors and deformation mechanisms was systematically investigated.

Figure 4a presents the true stress–true strain curves of the α_p_,_s_ + β Ti-45551 alloy at different strain rates at room temperature. Obviously, the dynamic true stress–true strain curves of the Ti-45551 alloy exhibited some oscillations due to the stress wave reflection between the surfaces of specimens and incident bar. Figure 4a presents the yield strength of the α_p_,_s_ + β Ti-45551 alloy, which increased by 240 MPa as the strain rate increased from 1 × 10^−^^3^ s^−^^1^ to 4 × 10^3^ s^−^^1^ at room temperature. Furthermore, there was no obvious change in elongation and strain hardening capability.

The true stress–true strain curves of the lamellar α_s_ + β phase Ti-45551 alloy at a range of strain rates is shown in Figure 4b. As shown in Figure 4b, the yield stress of the lamellar α_s_ + β phase Ti-45551 alloy was about 1120 MPa at 1 × 10^−^^3^ s^−^^1^, lower than the yield strength of the α_p,s_ + β Ti-45551 alloy (1260 MPa). This difference in yield strength may come from a Hall–Petch relationship with the size of the lamellar α_s_ of the Ti-45551 alloy. Under quasi-static loading conditions, Figure 4b shows that the lamellar α_s_ + β phase Ti-45551 alloy exhibits a clear strain hardening capability. Moreover, the yield strength (σ_y_) and failure strain (ε_f_) were about 1120 MPa and 6.1% at room temperature. As shown in Figure 4b, the Ti-45551 alloy exhibited a higher yield strength and larger ductility, implying that the mechanical properties of the lamellar α_s_ + β phase Ti-45551 alloy exhibit a strong strain rate dependence. Furthermore, it should be noticed that the lamellar α_s_ + β phase Ti-45551 alloy shows a clear strain softening capability at high strain rate loading conditions. Similar rate dependences have been reported in Ti6Al4V and AM50 alloys [16]

The slope factor of the yield strength versus the logarithm of the strain rate is the strain rate sensitivity (SRS) of materials [17]. Moreover, SRS coefficients for plastic deformation under different strain rate loadings could be described as follows:(4)$$m=\frac{dln\sigma }{dln\dot{\varepsilon }}$$

Figure 5 shows the variation of SRS for the Ti-45551 alloy with different phase compositions or microstructures. The lamellar α_s_ + β phase Ti-45551 alloy shows a lower SRS coefficient than the α_p,s_ + β Ti-45551 alloy.

### 3.2. Phase Composition Effects on Deformation Modes

Figure 6 shows the basic deformation modes, including dislocation slips and deformation twinning, in the HCP α phase and BCC β phase during plastic deformation at room temperature [18]. When the HCP α phase deforms at room temperature, traditionally only two independent slip systems can be activated in the basal plane. However, the BCC β phase exhibits a total of forty-eight slip systems at room temperature, i.e., twelve (112) [111] type slip systems, twelve (110) [111] type slip systems, and twelve (123) [111] type slip systems [19]. For polycrystalline materials, at least five independent slip systems are needed for the homogenous plastic deformation of materials [20]. As a consequence, the ductility of dual-phase Ti alloys is majorly governed by the HCP α phase, which exhibits limited slip systems. However, under shock loading, deformation twinning can act as a complementary deformation mode in accommodating the local plastic deformation in HCP material. Deformation twinning with the type of <10–12> {10–11} compression twin and <10–11> {10–12} extension twin always accommodates the local plastic strain along the c-axis.

Focusing on the deformation mechanisms of Ti-45551 under different loading conditions, we characterized the deformation mechanisms of the Ti-45551 samples after tensile tests. Figure 7 shows the major deformation mode of the Ti-45551 alloy with different phase compositions at quasi-static loading. From the bright field TEM image, the major deformation mechanisms are dislocation slip in both the α_p_, _s_ + β and lamellar α_s_ + β phase Ti-45551 alloys because high density dislocation can be found in deformed areas. Figure 7a shows that a high dislocation density structure can be found in the α_s_ lamellae and α_s_/β interface. However, the dislocation density in the equiaxed α_p_ phase is very low due to their limited slip systems at room temperature. A clear dislocation pile up can be found in the equiaxed α_p_ phase, as shown in Figure 7b. The dislocations that nucleate and accumulate at the phase boundaries would be impeded and have their movement stopped by the α phase, which contributes to the high strength and strain hardening effect of Ti-45551 alloys. Moreover, the nonuniform distribution of dislocation density can easily introduce stress concentration in the α/β interface, and then result in the low ductility of Ti-45551. Figure 7c also shows a lot of dislocation tangles (high density dislocation structure), such as those distributed in the α/β interface and β grains. Moreover, the dislocation density is also very low in the lamellar α_s_ phase, as shown in Figure 7d.

Figure 8 shows the major deformation mode in both the α_p_, _s_ + β and lamellar α_s_ + β phase Ti-45551 alloys under dynamic loading. After tension tests, there was a lot of deformation twinning in the equiaxed α_p_ phase, as shown in Figure 8a,b. Most deformation twins appeared as plates with a thickness of about 100 nm and across the entire equiaxed α_p_ phase, which is expected due to the dynamic loading. During plastic deformation, the twin boundaries within the primary α particle produce more nucleation sites for dislocation and impede dislocation motion. Thus, the dislocation density in the primary α particle is high, as seen in Figure 8b. The higher strength of the α_p_,_s_ + β Ti-45551 alloy is attributed to the additional hindering effect of the twin boundaries on dislocation motion under dynamic loading.

For the lamellar α_s_ + β phase Ti-45551 alloy, the plasticity of the lamellar microstructure is still primarily controlled by dislocation slip. Compare with the equiaxed α_p_ phase, the width of the lamellar α_s_ was only about 30–60 nm, which largely limits the twin nucleation and propagation even under shock loading. However, it can be observed that the dislocation density was similar in the lamellar α_s_ and β phase during the dynamic deformation process, as seen in Figure 8a. High dense dislocations are stacked in the lamellar α_s_. During dynamic deformation, more dislocations are activated from α_s_/β boundaries and slip in the HCP lamellar α_s_. The lamellar α_s_ phase can also contribute more plastic deformation under shock loading. Therefore, lamellar α_s_ + β phase Ti-45551 can exhibit a large ductility under dynamic loading conditions.

### 3.3. Phase Composition Effects on Failure Modes

It is widely accepted that fractographic features are helpful to analyze the fracture behavior and related performance of the materials. Therefore, for understanding the effect of strain rate on the tensile fracture mechanism, the tensile fracture morphology of the α_s_ + β and lamellar α_s_ + β phase Ti-45551 alloy under quasi-static and dynamic loading conditions were carefully observed.

Figure 9a shows the fractography of the tensile α_s_ + β Ti-45551 specimen under quasi-static loading (the arrow represents the shear direction). Figure 9a shows a mix fracture mode of dimple and crack under quasi-static loading. The cracks are parallel or perpendicular to the shear direction in the fracture areas. The typical fracture surface of the tensile α_s_ + β Ti-45551 specimen under dynamic loading (4 × 10^3^ s^−1^) is shown in Figure 9b. The major failure mode was dimples. The depth of the dimples was much deeper than that under the quasi-static loading and without obvious cracking, as shown in Figure 9b. Therefore, the shear failure mechanism is the major fracture mode of the Ti-45551 specimens under both quasi-static and dynamic loading conditions. Compared with the failure modes of the α_s_ + β Ti-45551 specimen, the α_p_,_s_ + β Ti-45551 alloy showed less cracking under quasi-static loading and more uniform dimples under dynamic loading. Moreover, the fractography of the Ti alloy also proves that the α_p_,_s_ + β Ti-45551 alloy exhibits an excellent ductility, especially under shock loading.

## 4. Conclusions

The mechanical behaviors, deformation mechanisms, and facture behaviors of dual-phase Ti-45551 alloy with different phase compositions under quasi-static and dynamic loading conditions at room temperature were reported. The results from the tensile testing show that the lamellar α_s_ + β phase Ti-45551 alloy exhibits higher strain rate sensitivity. Under dynamic loading, the strength and ductility increase under a high strain rate. Microstructure observations reveal that the transition from dislocation slip to defamation twinning occurs in the equiaxed α_p_ phase of the α_p_,_s_ + β Ti-45551 alloy during dynamic plasticity. Moreover, the lamellar α_s_ phase can also contribute more plastic deformation through more dislocation activated in the α_s_ phase under shock loading. The analysis of the fracture morphology showed that ductile dimples along the shear direction is the major fracture mode under both quasi-static and dynamic loading. Ti-45551 samples show more and deeper dimples, which represent more uniform plastic deformation at a high strain rate.

## Figures and Tables

**Figure 1 materials-15-04068-f001:**
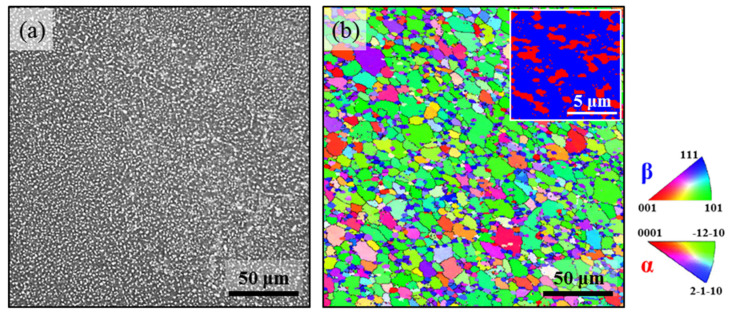
The microstructure of solution-treated Ti-45551 alloy: (**a**) SEM micrograph; (**b**) EBSD IPF map and phase map.

**Figure 2 materials-15-04068-f002:**
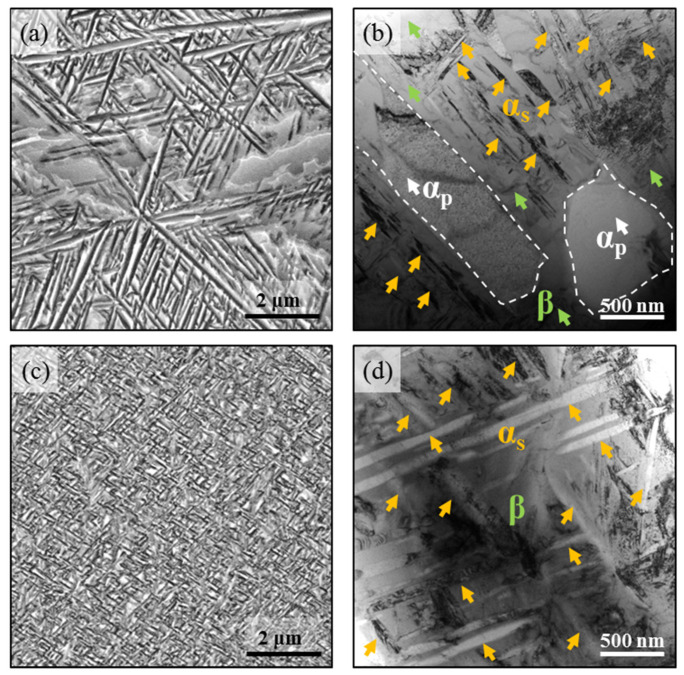
Typical microstructures of Ti-45551 alloy after different heat treatment. (**a**,**b**) Ti-45551 alloy after being aged at 360 °C for 8 h; (**c**,**d**) Ti-45551 alloy after being aged at 360 °C for 8 h + aged at 550 °C for 8 h. The yellow arrows represent αs, green arrows reprensent β and white arrows represent α_p_.

**Figure 3 materials-15-04068-f003:**
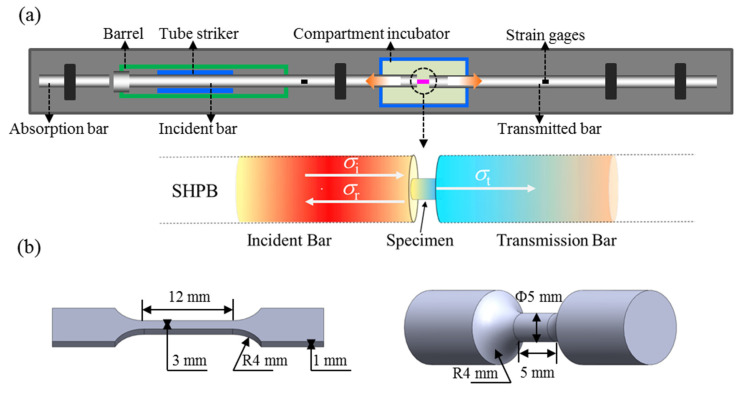
Schematic of split Hopkinson tensile bar (SHPB) and stress wave history during dynamic tension (**a**) and (**b**) the precise dimensions of the Ti alloy samples for quasi-static and dynamic tensile tests.

**Figure 4 materials-15-04068-f004:**
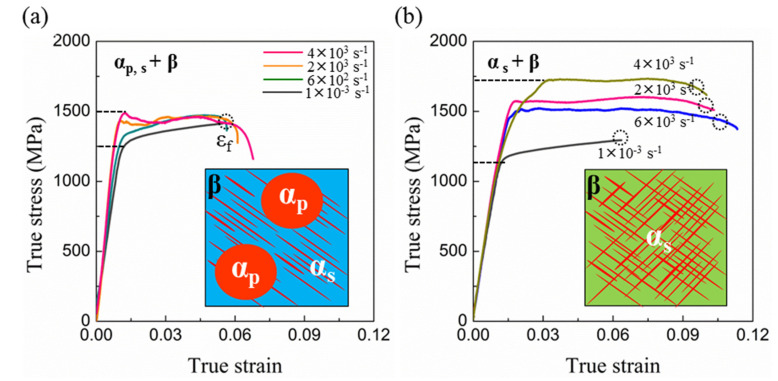
Mechanical behaviors of two kinds of α-β metastable β-Ti alloys at different strain rates. (**a**) Tensile stress–true strain curves of α_p,s_ + β Ti-45551 and (**b**) tensile true stress–true strain curves of α_s_ + β Ti-45551. The quasi-static curves (strain rate, 10^−3^ s^−1^) were obtained from MTS testing system and dynamic curves were obtained from SHPB (strain rate ~6 × 10^2^ s^−1^, 2 × 10^3^ s^−1^, and 4 × 10^3^ s^−1^).

**Figure 5 materials-15-04068-f005:**
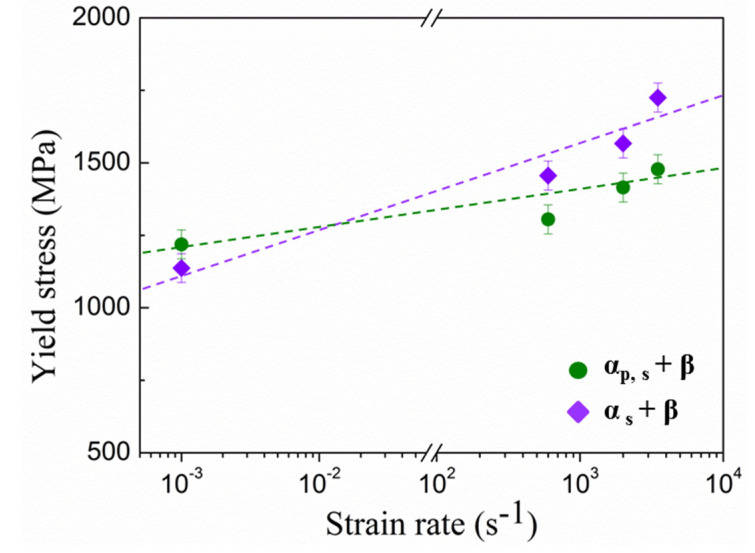
Yield strength of two kinds of α-β metastable β-Ti alloys at different strain rates. The quasi-static data (strain rate, 10^−3^ s^−1^) were obtained from MTS testing system and dynamic data were obtained from SHPB (strain rate ~6 × 10^2^ s^−1^, 2 × 10^3 s−1^, and 4 × 10^3^ s^−1^).

**Figure 6 materials-15-04068-f006:**
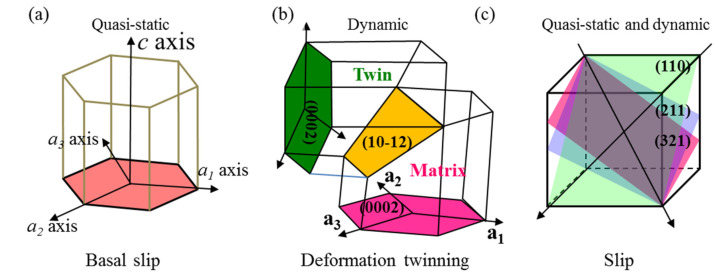
Schematic diagram of deformation modes in α-β Ti alloys. (**a**) HCP α phase at quasi-static loading; (**b**) HCP α phase at dynamic loading; (**c**) BCC β phase at quasi-static and dynamic loading.

**Figure 7 materials-15-04068-f007:**
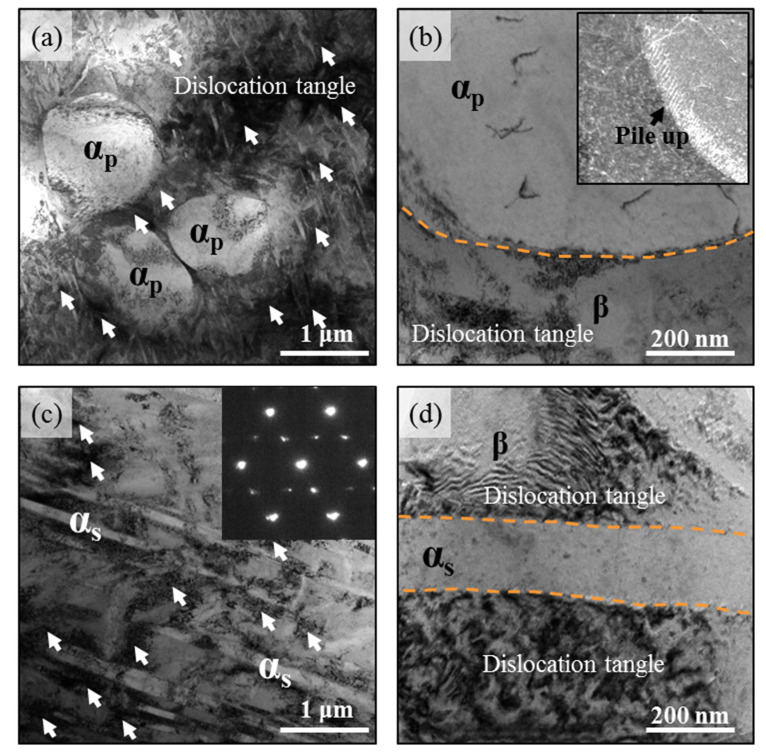
Deformation micromechanisms in two kinds of α-β metastable β-Ti alloys at quasi-static loading. (**a**) Bright field TEM image of deformed α_p,s_ + β Ti-45551 and its details (**b**); (**c**) bright field TEM image of deformed α_s_ + β Ti-45551 and its details (**d**). The white arrows represent dislocation tangles.

**Figure 8 materials-15-04068-f008:**
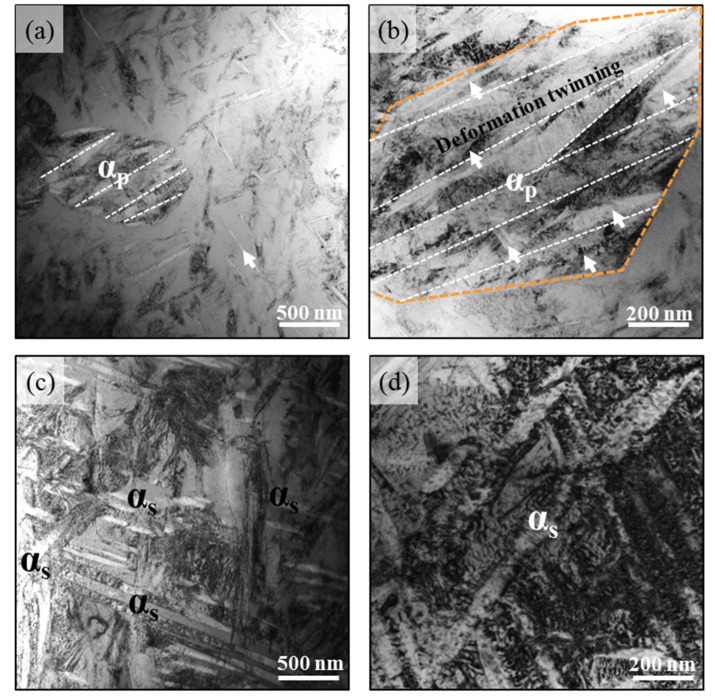
Deformation micromechanisms in two kinds of α-β metastable β-Ti alloys at dynamic loading. (**a**) Bright field TEM image of deformed α_p,s_ + β Ti-45551 and its details (**b**); (**c**) bright field TEM image of deformed α_s_ + β Ti-45551 and its details (**d**). The white arrows represent deformation twinning.

**Figure 9 materials-15-04068-f009:**
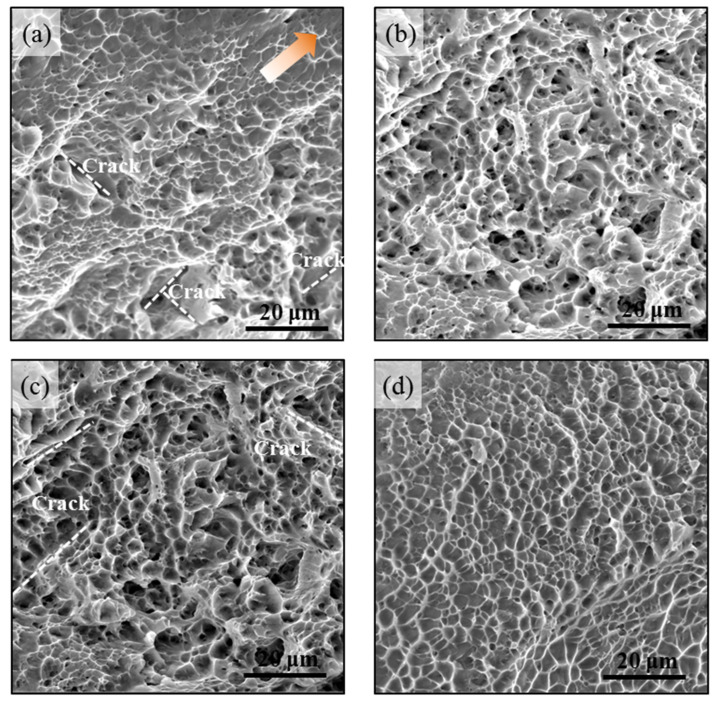
Fractography of α_s_ + β and α_p_,_s_ + β Ti alloy at quasi-static and dynamic loading conditions. The α_s_ + β Ti alloy under (**a**) quasi-static loading and (**b**) dynamic loading. The α_p_,_s_ + β Ti alloy under (**c**) quasi-static loading and (**d**) dynamic loading. The arrow represents the shear direction.

## Data Availability

Data sharing is not applicable to this article.

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
