# Peer review of "Phase Composition Effects on Dynamic Behavior and Strain Rate Sensitivity in Metastable β-Ti Alloys"

_materials, 2022, doi:10.3390/ma15124068_

Round 1

Reviewer 1 Report

The article is devoted to the study of the behavior of the Ti-45551 alloy under deformation at different strain rates. The article includes high-quality graphic material. Written clearly and competently. However, I have a few questions and comments for the authors of the paper:

  1. What type of tensile test pieces were used? What were the sample sizes? What type of machining was used? I propose to bring a photo or sketch of samples.
  2. An extensometer was used for tensile testing. Did the sample break in the area of ​​the extensometer or was the break located outside it?
  3. Samples for testing on the SHPB facility are not described. Not all readers may be so familiar with this research method.
  4. Figure 4 shows tensile stress-true strain curves. Are these curves obtained from tensile tests and from the SHPB? Maybe it's worth specifying which values ​​are obtained during the tensile test, and which during the SHPB? Also, in Figure 5.
  5. Why are the experimental points approximated by a straight line in Figure 5?
  6. From what part of the samples were microsections made?
  7. Which samples were fractographical analyzed? Why is the fractographic analysis of αp,s+β Ti 45551 not shown?

Author Response

Dear reviewers,

  We would like to express our sincere appreciation for your careful reading and invaluable comments to improve our manuscript- Phase composition effects on dynamic behaviour and strain rate sensitivity in metastable β-Ti alloys. We have substantially revised our manuscript after reading these invaluable comments provided by the reviewers. Below we outline the point-to-point responses to the reviewer’s comments. All modifications are highlighted in yellow below and in the revised manuscript. 

Response to Reviewer #1

The article is devoted to the study of the behavior of the Ti-45551 alloy under deformation at different strain rates. The article includes high-quality graphic material. Written clearly and competently. However, I have a few questions and comments for the authors of the paper:

Question 1: What type of tensile test pieces was used? What were the sample sizes? What type of machining was used? I propose to bring a photo or sketch of samples.

Response: Very thanks for your good question. Of course, there exists difference in sample size and type under different loading conditions-quasi-static and dynamic. In the revised version, we give the specific dimensions of tensile samples.

The precise dimensions of the Ti alloy samples for quasi-static and dynamic tensile tests are shown in Fig. 3(b).

Figure 3. Schematic of split Hopkinson tensile bar (SHPB) and stress wave history during dynamic tension (a) and The precise dimensions of the Ti alloy samples for quasi-static and dynamic tensile tests.   

Question 2: An extensometer was used for tensile testing. Did the sample break in the area of ​​the extensometer or was the break located outside it?

Response: This question is focus on the experimental details. During quasi-static, the samples break in the area of the extensometer. And we revised the description of our quasi-static tests.

During the tensile test, the elongation is recorded by an extensometer with gauge length of 12 mm. As tensile samples break in the working area of the extensometer, the total strain histories are recorded by the testing systems under quasi-static loading.

Question 3: Samples for testing on the SHPB facility are not described. Not all readers may be so familiar with this research method.

Response: The precise dimensions of samples, especially for samples for testing on the SHPB facility, we give the size and type in Fig. 3 in our revised version. And you first question has mentioned this.

The precise dimensions of the Ti alloy samples for quasi-static and dynamic tensile tests are shown in Fig. 3(b) and (c), respectively.

Question 4: Figure 4 shows tensile stress-true strain curves. Are these curves obtained from tensile tests and from the SHPB? Maybe it's worth specifying which values ​​are obtained during the tensile test, and which during the SHPB? Also, in Figure 5.

Response: This is a really good question. The quasi-static curves obtained from MTS testing system (strain rate 10-3 s-1) and the dynamic curves obtained from SHPB (strain rate ~6 × 102 s−1, 2 × 103 s−1 and 4 × 103 s−1). And we revised the corresponding figure captions of Fig. 4 and 5.

Figure 4. Mechanical behaviors of two kinds of α-β metastable β Ti alloys at different strain rate. (a) Tensile stress- true strain curves of αp,s+β Ti 45551; (b) Tensile true stress- true strain curves of αs+β Ti 45551. The quasi-static curves (strain rate, 10-3 s-1) obtained from MTS testing system and dynamic curves obtained from SHPB (strain rate ~6 × 102 s−1, 2 × 103 s−1 and 4 × 103 s−1).

Figure 5. Yield strength of two kinds of α-β metastable β Ti alloys at different strain rates. The quasi-static data (strain rate, 10-3 s-1) obtained from MTS testing system and dynamic data obtained from SHPB (strain rate ~6 × 102 s−1, 2 × 103 s−1 and 4 × 103 s−1).

Question 5: Why are the experimental points approximated by a straight line in Figure 5?

Response: When we investigated the relationship between strength and strain rates, a linear approximation is obtained. And the slop of this linear fitting (straight line) is defined as strain rate sensitivity (SRS), m. And we add this description in our revised version.

The slop factor of the yield strength versus the logarithm of the strain rate is strain rate sensitivity (SRS) of materials [17]. And SRS coefficients for plastic deformation under different strain rate loading could be described as:

     (4)

Question 6: From what part of the samples were microsections made?

Response: This is a good question. After plastic deformation, the TEM specimens for microstructure characterizations are cut from deformed regions, which are near the fracture positions. In order to make it clear to other readers, we have revised the description part in 2.3 microstructure characterization.

After plastic deformation, the TEM samples were taken from the deformed region. The bright field and the high resolution images of the deformed Ti alloy were acquired by JEOL F200. TEM disk samples were used mechanical polishing and thinned to perforation precision ion thinning treatment.

Question 7: Which samples were fractographical analyzed? Why is the fractographic analysis of αp,s+β Ti 45551 not shown?

Response: After quasi-static (strain rate 10-3 s-1) and dynamic loading (4 × 103 s−1), the αs+β Ti and αp, s + β Ti-45551 alloy samples were fractographical analyzed. And the failure modes are quite similar. We add the fractographical analysis of αp, s + β Ti-45551 alloy samples in our revised version. Many thanks for this question.

Therefore, for understanding the effect of strain rate on tensile fracture mechanism, tensile fracture morphology of the αs+β and lamellar αs+β phase Ti-45551 alloy under quasi-static and dynamic loading conditions are carefully observed.

Compare with the failure modes of the αs+β Ti-45551 specimen, the αp, s + β Ti-45551 alloy shows less crack under quasi-static loading and more uniform dimples under dynamic loading. And the fractography of Ti alloy also prove that the αp, s + β Ti-45551 alloy exhibit excellent ductility especially under shock loading. 

Reviewer 2 Report

The presented paper is well written, investigations are well planned and presented.  

In my opinion, the suggestion presented below could only improve the quality of the work:

1) There is no EBSD study of material after loading, it would be intersting.

2) TEM diffraction patterns from both phases are recommended to confirm their present in the marked area. 

3) The calculation of the SRS coefficients should be presented and the values of them. 

Author Response

Dear reviewers,

  We would like to express our sincere appreciation for your careful reading and invaluable comments to improve our manuscript- Phase composition effects on dynamic behaviour and strain rate sensitivity in metastable β-Ti alloys. We have substantially revised our manuscript after reading these invaluable comments provided by the reviewers. Below we outline the point-to-point responses to the reviewer’s comments. All modifications are highlighted in yellow below and in the revised manuscript. 

Response to Reviewer #2

The presented paper is well written, investigations are well planned and presented. 

In my opinion, the suggestion presented below could only improve the quality of the work:

Question 1: There is no EBSD study of material after loading, it would be interesting.

Response: Thank you for your good suggestion. In fact, we also want to give high quality EBSD images of specimens after quasi-static and dynamic loading. However, after loading the resolution ratio is very low due to the residual stress. And the width of αs is about 30-60 nm and our EBSD step size is about 100 nm. Of course, if we can perform EBSD study of material after loading, it would be very useful information for deformation mechanisms investigating. We will try that in the future work. 

Question 2: TEM diffraction patterns from both phases are recommended to confirm their present in the marked area.

Response: This is a very good question. And we add the diffraction patterns in Figure 7.

Again, we express our highest gratitude to the reviewer for the valuable comments and his/her efforts which enable us to improve our manuscript.

Round 2

Reviewer 1 Report

The authors have made additions to the article, in accordance with my comments. I recommend publishing the article in the journal in an updated version. Unfortunately, during the secondary review, the article is not presented in the template.